







# Human influences on streamflow drought characteristics in England and Wales

Erik Tijdeman[1], Jamie Hannaford[2], Kerstin Stahl[1]

[1] Faculty of Environment and Natural Resources, University of Freiburg, Freiburg, Germany
[2] Centre for Ecology and Hydrology, Wallingford, UK

*Correspondence to*: E. Tijdeman (erik.tijdeman@hydrology.uni-freiburg.de)

**Abstract.**

Human influences can affect streamflow drought characteristics and propagation. The question is where, when and why? To answer these questions, the impact of different human influences on streamflow droughts were assessed in England and Wales, across a broad range of climate and catchments conditions. We used a dataset consisting of catchments with near-natural flow as well as catchments for which different human influences have been indicated in the metadata ('Factors Affecting Runoff') of the UK National River Flow Archive (NRFA). A screening approach was applied on the streamflow records to identify human influenced records with drought characteristics that deviated from those found for catchments with near-natural flow. Three different deviations were considered, specifically deviations in: 1) the relationship between streamflow drought duration and the Base Flow Index; 2) the correlation between streamflow and precipitation and 3) the temporal occurrence of streamflow droughts compared to precipitation droughts, i.e., an increase or decrease in streamflow drought months relative to precipitation drought months over the period of record. The identified deviations were then related to the indicated human influences. Results showed that the majority of catchments for which human influences were indicated did not show streamflow drought characteristics that deviated from those expected under near-natural conditions. For the catchments that did show deviating streamflow drought characteristics, prolonged streamflow drought durations were found in some of the catchments affected by groundwater abstractions. Weaker correlations between streamflow and precipitation were found for some of the catchments with reservoirs, water transfers or groundwater augmentation schemes. An increase in streamflow drought occurrence towards the end of record was found for some of the catchments affected by groundwater abstractions and a decrease in streamflow drought occurrence for some of the catchments with either reservoirs or groundwater abstractions. In conclusion, the proposed screening approaches were successful in identifying streamflow records with deviating drought characteristics that are likely related to different human influences. However, a quantitative attribution of the impact of human influences on streamflow drought characteristics requires more detailed case by case information about the type and degree of all different human influences. Given that, in many countries, such information is often not readily accessible, the approach adopted here could provide useful in targeting future efforts. In England and Wales specifically, the catchments with deviating streamflow drought characteristics identified in this study could serve as the starting point of detailed case study research.





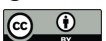

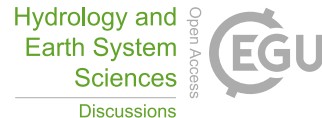

## 1 Introduction

Droughts pose a threat to water security all around the world. They are monitored with a variety of drought indices that represent different domains of the hydrological cycle and on a variety of scales (Bachmair et al., 2016). Meteorological drought indices are popular components of such drought monitoring systems, but in reality drought impacts are primarily

caused by deficits in other domains of the hydrological cycle, e.g. soil moisture, groundwater, streamflow; as demonstrated for the recent high-impact 2015 European drought (Van Lanen et al., 2016). Meteorological drought indices may not always sufficiently represent the hydrological situation on the ground for a variety of reasons related to natural catchment processes (summarized in the review of Van Loon, 2015). In addition to natural processes, human influences in river catchments, such as abstractions or reservoir operations, can intensify or mitigate hydrological droughts (Van Loon et al., 2016b).

Furthermore, human influences related to management practices such as changes in minimum flow requirements, increased abstraction of groundwater or river restoration programs may affect the occurrence of low flow periods over the period of record.

In a human-modified world, understanding when, where, why, and to what degree, different human influences have modified streamflow drought propagation and characteristics in the past is useful for the development of drought management and

mitigation strategies and is pivotal context for interpreting future streamflow drought projections and scenarios. Furthermore, from a drought monitoring and early warning perspective, it is important to understand how streamflow records have changed due to various human influences; primarily because non-stationarity in low flows (such as trends or step changes) potentially hinders the suitability of streamflow drought indices, which are often expressed relative to the historical record, to adequately represent actual drought events consistently through time. For example, an increase in minimum flow

requirements could moderate streamflow drought severity over time, limiting the potential of a streamflow drought index to detect droughts relative to the previous record; although, from a management perspective, the relative lack of streamflow droughts is a real effect. Conversely, increasing abstraction rates might result in the identification of more persistent or severe streamflow droughts. Again, this is a real effect from the point of view of streamflow drought, even if from a water supply perspective the action serves to moderate drought impacts. These complexities have consequences for the

interpretation of hydrological drought indices, i.e., they still reflect droughts and impacts related to the instream flow environment (e.g. impacts on aquatic ecosystems), however, they might fail to adequately describe other drought related impacts (or falsely indicate 'drought' during wet conditions).

In order to identify human influenced streamflow records and achieve a better understanding of how human influences affect drought characteristics, these influences must first be separated from natural controls. In catchments free of human

influences, natural controls related to meteorology and catchment characteristics determine the onset, duration and termination of drought events. Defined relative to a normal seasonal streamflow, the onset of a streamflow drought is generally caused by an anomaly in meteorological variables such as below normal precipitation or above normal temperatures causing higher evaporation, or by storage of any precipitation input e.g. as snow (Van Loon & Van Lanen,



2012). The persistence of a streamflow drought and its characteristics, such as duration and severity, are influenced by the combination of meteorological anomalies and catchment characteristics. Barker et al., (2016) quantified the influence of catchment precipitation and storages (indexed by the Base Flow Index, BFI, and also numerous catchment properties) on the median and maximum drought duration for a dataset of near-natural catchments in the UK. For Austria, Van Loon & Laaha

(2015) found that the main control on drought duration was related to catchment processes (represented by the BFI). Tijdeman et al. (2015) tested these controls, amongst others, on a dataset of catchments with near-natural flow in Europe and the USA and found that the duration of the more extreme streamflow drought events was higher in classes of catchment with a high BFI. For drought termination, a surplus of water that compensates for the accumulated deficit is important. Within the UK, streamflow drought termination duration is correlated with elevation and catchment precipitation; i.e. droughts tend to

terminate more abruptly in wetter upland areas which are less permeable (Parry et al., 2016).

Together with natural controls, human influences, such as reservoir operations and abstractions, can influence the onset, duration and termination of streamflow drought events. These human influences can both intensify or mitigate streamflow droughts compared to the natural situation. However, it is often difficult to isolate these human influences from the natural ones. The framework provided by Van Loon et al., (2016a) suggests different approaches to investigate human influences on

streamflow drought.

A first approach is to compare meteorological droughts with streamflow droughts. This approach is relatively straightforward and only relies on the availability of meteorological time series. However, it does not account for the fact that flows in different catchments are sensitive to meteorological deficits over different timescales (e.g., Haslinger et al, 2014; Barker et al., 2016). Another approach is based on a comparison between the influenced and non-influenced part of the

record for a particular location. Such comparisons are commonly done in the field of ecohydrology to quantify the effects of a known impact (e.g. dam construction) using pre- and post-impact time series. For example, Richter et al., (1996) proposed a methodology for comparing natural and influenced records using 32 properties of the hydrological regime including annual low flows and durations of below threshold flow. Alternatively, baseline or naturalised flow regimes may be constructed using a model calibrated on the pre-impact time series to replicate natural flow conditions during the period of impact (e.g.,

Van Loon & Van Lanen, 2013). This approach has the advantage that comparisons between 'pre-impact' and 'post-impact' are influenced by climatological variability, whereas modelled series can be constructed for the same period of climate forcing. However, the above described approaches are predicated on the availability of 'pre-impact' time series, or more generally some period where human influences are not present. Pre-impact series from before impoundments were constructed are generally rare, and they are especially rare for more diffuse impacts such as abstractions. An alternative

approach is based on the principles of a paired catchment analyses, a concept that has been a foundation of process hydrology. Typically, a paired catchment study compares the flow regimes of nearby catchments with similar physical characteristics. The approach has been applied in numerous iconic experimental studies to investigate land use impacts on river flow (e.g. review of Brown et al., 2005). However, the paired catchment concept can also be used to study human influences on streamflow, using existing gauging station networks, if appropriate 'donor' natural catchments with similar



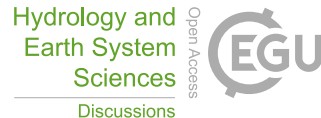

flow regimes can be found for 'target' catchments with known influences (as conducted in the case of urbanisation effects on floods; Prosdocimi et al., 2015). For drought research, several studies use a variation of this approach to investigate the impact of reservoirs on streamflow droughts by comparing (undisturbed) upstream records with downstream records before and after the construction of the reservoir (e.g. López-Moreno et al., 2009; Wen et al., 2011; Rangecroft et al., 2016).

Most research on human influences on streamflow drought characteristics and propagation has been carried out as case studies or with modelled data. Studies that assess human impacts on streamflow drought propagation based on observed streamflow drought characteristics at larger regional to national scales, across a broad range of catchment types, are less common. This study aims to close this gap and, following the recommendation of Barker et al., (2016), seeks to understand the human influences on streamflow droughts in England and Wales, densely populated regions with a long settlement

history and thus prevalent human influences on river flow. For such a national scale assessment, it is not feasible to obtain all information and explicitly consider all the different types and degrees of human influences (including management practices that often change over time). Rather, here we propose a 'screening' approach that seeks to identify impacts in a large network of catchments, through identifying streamflow characteristics that deviate from what may be expected under 'natural' conditions (similar to, e.g., Carlisle et al., (2011) for flow magnitude changes in the US or Sadri et al., (2016) for

low flows in the Eastern US). This study used a diverse dataset of nearly 200 streamflow (and precipitation) records, including a range of known potential human influences. The approach is based on screening for catchments with identifiable deviations from 'natural' conditions on the basis of 1) streamflow drought duration, 2) correlation between precipitation and streamflow and 3) temporal changes in streamflow drought occurrence.

## 2 Study Area and Data

England and Wales have very diverse climate characteristics and catchment properties, and the availability of freshwater and total water demand varies significantly across the region (Acreman, 1999). There is a large gradient in annual average precipitation (more precipitation in the North-West and less in the South-East) and water demand varies, with some of the highest demands being in the drier south and east where there are major urban centres and concentrations of intensive agriculture. Many rivers in England and Wales have been modified substantially since at least Roman times. Since the

industrial revolution, there have been large-scale interventions to secure freshwater supply, affecting the hydrological properties of rivers across the region. Reservoirs were constructed for e.g., public water supply, flood control or hydropower production but have later on also been used to maintain good environmental flow quality (Acreman et al., 2009). Water supply from groundwater abstractions mainly occurs in densely populated lowland areas in England and Wales from several major aquifers, most notably from the Cretaceous Chalk. These aquifers have been over-exploited in the past, and some

more recent management practices aim to reduce these effects, especially during situations of low flow. Other management practices include water transfers (often crossing catchment boundaries) and augmentation of flow with effluent return, where the source of the effluent water is not necessarily from within the catchment.





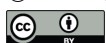

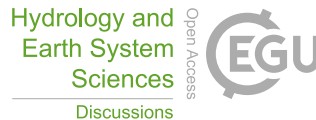

Streamflow time series used in this study stem from the National River Flow Archive (Dixon et al., 2013; http://nrfa.ceh.ac.uk/). Stations were selected based on the longest common total period of recent daily data availability, resulting in the period 1974-2013, with gaps in streamflow records of a maximum of 5 days per calendar month of missing data allowed. This is a fairly strict missing data criterion, limiting the number of available catchments. However, a strict

criterion is necessary when calculating streamflow drought event characteristics. Infilling using near-neighbour or other analogues (e.g., Harvey et al., 2012) was not deemed appropriate given the inclusion (by design) of many heavily human-modified catchments. The resulting dataset consisted of 187 catchments. Monthly precipitation time series for these catchments were extracted from the NRFA catchment monthly rainfall series for the time span 1973-2013.

Metadata consist of information on both natural controls and human influences that are hypothesized to influence streamflow

drought characteristics. One of the most important natural controls on streamflow drought is catchment storage. To index this, we use a variation of the Base Flow Index (BFI) that is predicted from properties of soil (HOST, acronym standing for Hydrology Of Soil Types) classes, the BFIHOST classification (Boorman et al., 1995). BFIHOST was preferred over the BFI calculated directly from the streamflow record, which would lead to circularity as the latter could itself be altered by human influences present in the record. The BFIHOST has been widely used in the UK to define catchment similarity, in a

host of regionalisation methods (see for examples Hannaford et al., (2013) and references therein).

Information about human influences is pivotal to this kind of study. In the UK, information on the nature and extent of human influences is provided by the NRFA. A readily-accessible categorical scheme is the Factors Affecting Runoff (FAR) classification, which is presented on the website of the NRFA (http://nrfa.ceh.ac.uk/catchment-summary-information) and in the UK Hydrometric Register (Marsh & Hannaford, 2008). FAR are one letter codes that represent the presence of different

types of human influences. There are 8 different FAR, 7 indicating different types of human influences and one indicating near-natural flow (Table 1). These near-natural flags are distinct from the UK Benchmark Network classification, a designated network of near-natural catchments (Harrigan et al., *submitted*). For internal consistency, we have used the 'N' flag from the FAR classification rather than Benchmark status; the latter is a more complex, multi-criteria definition of natural catchments, with other criteria including low flow hydrometric performance, record length, spatial

representativeness, etc.

Each catchment can have multiple FAR codes, depending on the number of human influences. It should be noted that these codes are simple presence/absence indicators of such influences, and are therefore purely indicative of possible impacts. The FAR codes are supported by qualitative information in the NRFA's thumbnail station descriptions provided for each site. The FAR codes do not guarantee that an impact is detectable, nor do they enable quantification of the scale of impact.

Quantitative information on impacts is more limited; information on abstractions and discharges are held by the regulators, but this is not widely accessible on a routine basis. Some studies have employed modelled estimates based on this information (Hannaford et al, 2013), but those data were not used here owing to the uncertainties in modelling and their focus on low flows (Q95) rather than streamflow drought. Moreover, here we are advocating a rapid assessment 'screening' methodology that could be used in other contexts or other countries. The benefit of the FAR approach is that simple





presence/absence codes are relatively straightforward to apply in other contexts, unlike quantitative assessments of human impacts on flow regimes.

For this study, we mainly focus on catchments with near-natural flow records (FAR=N), groundwater abstractions (FAR=G) and storage or impounding reservoirs (FAR=S). We focus on these human influences because we hypothesize they

potentially exert the largest influence on the flow record (and correspondingly also on streamflow drought characteristics). Furthermore, in the UK, groundwater abstractions have often been related to decreasing low flow (e.g., NRA., 1993) whereas storages or impounding reservoirs have been shown to alter the entire flow regime (e.g., Acreman et al., 2009). Note that a catchment can have other FAR, additional to the ones mentioned above, e.g., groundwater abstractions can be accompanied by FAR indicating direct river abstraction for public water supply. Station locations and their classifications

according to the considered FAR are presented in Fig. 1.

## 3 Methods

### 3.1 Drought characteristics

For all catchments, monthly average streamflow series $Q(t)$ were derived for each calendar month $t$ within the considered time period (1974-2013). Monthly accumulated precipitation $P(t)$ for each of these catchments was summed over 12

different accumulation periods in a similar way as is done for the Standardised Precipitation Index (SPI); for each month $t$ within the considered time period 12 different precipitation values ($P_n(t)$, $n=1 \dots 12$) were derived ranging from $P_1(t)$, the precipitation in the current month, to $P_{12}(t)$, the precipitation in the current month and the 11 previous months.

Droughts were then identified from both $Q(t)$ and $P_n(t)$ (hereafter, $x(t)$ is used when referring to either of the two hydro-meteorological variables) using a monthly variable threshold level approach. Such a monthly variable threshold method

takes into account the natural variability of $Q$ and $P_n$, and reflects that the impact of human influences might be related to a certain time of the year (Van Loon et al., 2016a). For each calendar month of $x(t)$, a threshold level ($\tau_x$) was defined based on the 20[th] percentile of $x$ for that calendar month. Similar to Tallaksen et al, (2009), we created a binary index time series $I_x(t)$ that specifies for each monthly time step $t$ if hydro-meteorological variable $x(t)$ is at or below the threshold $\tau_x(t)$ and thus a streamflow or precipitation anomaly below the threshold, here for simplicity termed 'drought' (Eq. 1, example in Fig. 2).

$$I_x(t) = \begin{cases} 1 \text{ if } x(t) \leq \tau_x(t) \\ 0 \text{ if } x(t) > \tau_x(t) \end{cases} \tag{1}$$

Then, the combined durations $L_{T,x}$ of each drought event $j$ were computed for each catchment and hydro-meteorological variable $x$ (Eq. 2).


$$L_{T,x}[j] = \sum_{t=1}^{L_x[j]} I_x \tag{2}$$



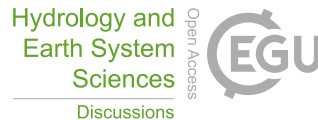

Where $L_x[j]$ is the duration of drought event j. For the example in Fig. 2, $L_{T,x}[j] = 9$ months. The following two characteristics were used in the continuation of this study:

1) average drought duration ($\overline{L_{T,x}}$),
2) the relative cumulative sum of drought occurrence ($C_x$); which is the cumulative sum of $I_x$ at each monthly time step (t = 1 … 480) divided by the total sum of $I_x$,

$C_x$ is formulated as a fraction of its maximum and reflects the relative cumulative sum of monthly drought occurrence ranging between 0 and 1. The shape of $C_x$ reveals whether monthly drought occurrence or deficit volume was equally distributed over the period of record or whether a larger proportion of monthly drought occurrence appears in the beginning, middle or towards the end of the record (exemplified in Fig. 3).

**3.2 Separating human influences from natural controls**

To link the alteration of streamflow drought characteristics to human influences, one approach is to characterise the deviation of streamflow drought characteristics from those expected under 'natural' conditions. These natural conditions can be represented by the driving precipitation drought or by near-natural flow records. The fundamental premise is to compare the variability in the relationship between streamflow drought characteristics and climate or catchment characteristics for catchments with near-natural flow (baseline) with the relationships observed for catchments with various human influences. Deviations from the expected relationship under near-natural conditions can potentially be attributed to human influences. For catchments with no or minimal human influences, there is typically a strong relationship between streamflow drought duration and climate and catchment properties (previously shown for the relation between duration and the BFI(HOST) by Van Loon & Laaha 2015; Barker et al., 2016; Tijdeman et al., 2016). Furthermore, for catchments with no or minimal human influences, streamflow drought indices are strongly correlated with meteorological drought indices (shown for the UK in Barker et al., 2016). The latter study showed that the accumulation period of the meteorological drought index with the highest correlation with streamflow is dependent on catchment characteristics; for catchments with substantial natural storage (e.g., groundwater-fed catchments) a higher correlation was found between streamflow and long-term precipitation (e.g., P12) whereas for impermeable catchments, a higher correlation was found between streamflow and short-term precipitation deficits (e.g., P1). Following a similar reasoning, the relative cumulative sum of monthly streamflow drought occurrence ($C_Q$) is expected to be related to the relative cumulative sum of long-term precipitation drought occurrence (e.g., $C_{P12}$) for slow responding (groundwater-fed) catchments and to short term precipitation drought occurrence (e.g., $C_{P1}$) for impermeable (responsive) catchments.

We defined three hypotheses related to expected deviations from the near-natural case caused by human influences:

$H_1$: The relation between $\overline{L_{T,Q}}$ and BFIHOST for catchments with human influences differs from this relation for catchments with near-natural streamflow records.



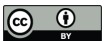



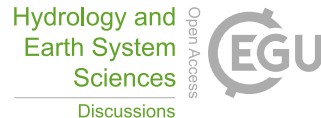

$H_2$: The maximum correlation between Q and Pn is lower for catchments with human influences compared to this maximum correlation for catchments with near-natural flow.

$H_3$: The minimum difference in cumulative temporal drought occurrence distribution of streamflow ($C_Q$) and precipitation ($C_{Pn}$) is larger for catchments with human influences than for catchments with near-natural flow.

$H_1$ was tested by graphically comparing the relation between BFIHOST and $\overline{L_{T,Q}}$ for catchments with near-natural flow records (FAR=N) and catchments with various human influences (FAR = G, R or other). We used a 95% confidence ellipse of the stations with near-natural flow as a baseline to define the expected range in relation between BFIHOST and $\overline{L_{T,Q}}$. To further test whether this relationship was independent of precipitation drought occurrence, we graphically compared how

10   much these characteristics were amplified compared to same characteristics of P1 and P12 (i.e., $\overline{L_{T,Q}} / \overline{L_{T,P1}}$ and $\overline{L_{T,Q}} / \overline{L_{T,P12}}$).

$H_2$ was tested by computing Spearman's rank correlation ($\rho$) between Q and Pn (n=1…12) for each catchment and calendar month. The $5^{th}$ quantile of the maximum rank correlation between Q and Pn ($\rho_{max}$) for catchments with near-natural flow was

15   used as a baseline; lower correlations were identified as potentially attributable to human influences.

$H_3$ was tested by comparing the temporal distribution of monthly streamflow drought occurrence ($C_Q$) with the temporal distribution of precipitation drought occurrence ($C_{Pn}$) for each catchment. The absolute difference ($A_{dif}$) was calculated for each combination of $C_{Pn}$ and $C_Q$ following Eq. 3.

$$A_{dif}[n] \ = \ C_Q - C_{Pn} \tag{3}$$

For n = 1 … 12. In Fig. 3, the area between two lines exemplifies $A_{dif}$. The minimum of $A_{dif}$ ($A_{dif,min}$) was used as a measure to reflect how well the temporal distribution of monthly precipitation drought occurrence relates to the temporal

25   distribution of monthly streamflow drought occurrence. The variability of $A_{dif,min}$ under near-natural conditions was again used as a baseline to identify catchments with deviating streamflow drought occurrence distributions; the $95^{th}$ quantile of $A_{dif,min}$ of streamflow records with near-natural flow serves as a baseline. For the subgroup of stations with a larger $A_{dif,min}$, $C_x$ were further examined graphically.





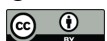

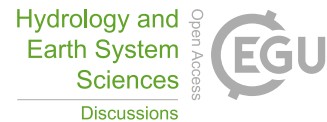

## 4 Results

### 4.1 Drought characteristics

Figure 4 shows the relation between average streamflow drought duration ($\overline{L_{T,Q}}$) and BFIHOST for catchments with near-natural flow records (FAR=N) and for catchments affected by different human influences. The catchments with near-natural
flow show a more or less linear relationship between $\overline{L_{T,Q}}$ and BFIHOST; longer streamflow droughts for slower responding catchments (higher BFIHOST). Some of the catchments affected by groundwater abstractions show a similar linear relationship between BFIHOST and $\overline{L_{T,Q}}$; however, the remaining catchments show $\overline{L_{T,Q}}$ that deviate from this linear relationship. $\overline{L_{T,Q}}$ is higher for these catchments (especially for catchments with a higher BFIHOST) and a substantial proportion of the points are located outside the confidence ellipse of catchments with near-natural flow.

Figure 5 reveals the amplification of average streamflow drought duration compared to average precipitation drought duration accumulated over a one-month period ($\overline{L_{T,Q}}$ / $\overline{L_{T,P1}}$, upper row) and accumulated over a 12-month period ($\overline{L_{T,Q}}$ / $\overline{L_{T,P12}}$, lower row). The patterns in Fig. 4 and 5 are comparable. Some of the catchments affected by groundwater abstractions again show a non-linear relation between $\overline{L_{T,Q}}$ / $\overline{L_{T,P1}}$ and BFIHOST. $\overline{L_{T,Q}}$ / $\overline{L_{T,P1}}$ is always larger than one, meaning that average streamflow drought duration is always higher than average monthly precipitation drought duration.
This is not the case when average streamflow drought duration is compared to average drought duration of long-term (12-month) precipitation records. $\overline{L_{T,Q}}$ / $\overline{L_{T,P12}}$ is often smaller than one (maximum 1.05 for catchments with FAR=N). However, some catchments with FAR=G that have a higher BFIHOST show a larger $\overline{L_{T,Q}}$ / $\overline{L_{T,P12}}$ (ranging between 1 and 2.86). An example of a catchment with a high average streamflow drought duration ($\overline{L_{T,Q}}$ = 8.73, $\overline{L_{T,Q}}$ / $\overline{L_{T,P12}}$ = 1.82) is the river Mimram at Panshangers Park (shown in S1).

### 4.2 Correlation between precipitation and streamflow

Figure 6 reveals the maximum correlation between Q and Pn ($\rho_{max}$) for the different calendar months. For catchments with FAR=N, the maximum correlation between streamflow and precipitation is generally strong; median of $\rho_{max}$ ranges between 0.92 (December) and 0.84 (April), upper bound 90% range between 0.97 (December) and 0.91 (May) and lower bound 90% range between 0.82 (October) and 0.69 (April). A percentage of stations with FAR≠N show a $\rho_{max}$ below the 90% range of
$\rho_{max}$ of stations with near-natural flow (ranging between 6% in April and 38% in September); however, differences are often small. Few catchments (n=9) show a $\rho_{max}<0.5$ for one or more calendar months, mostly in summer months. From these nine catchments, five are labelled with FAR=S (Fig. 6, panel a, f, g, h, i), which indicates the presence of storage or impounding reservoirs. In these cases, releases from reservoirs for the compensation of low flow downstream are the likely cause of the diminished correlation between precipitation and meteorology (example shown in S2). The other four catchments with
$\rho_{max}<0.5$ have FAR that indicate, amongst other factors, groundwater abstractions. For two cases (Fig. 6, panel d & e), flow augmentation by groundwater abstraction during drought is the likely cause of the lower maximum correlation between





precipitation and streamflow (example is presented in S3). In another case (Fig. 6, panel b), water transfers are likely to diminish the correlation between streamflow and precipitation (example is shown S4).

### 4.3 Temporal distribution of drought

Figure 7 presents the minimum absolute difference ($A_{dif,min}$) between the relative cumulative sum of drought occurrence of
streamflow ($C_Q$) and precipitation ($C_{Pn}$). A majority of the catchments (including most catchments with near-natural flow records) show comparable $A_{dif,min}$ (independent of the BFIHOST). The 95$^{th}$ quantile (baseline) of $A_{dif,min}$ of catchments with near-natural flow records is 24.3; for graphical inspection of $C_{Pn}$ and $C_Q$, we focus on the subgroup of 23 catchments that have a larger $A_{dif,min}$ (>24.3, Fig. 8).

For this selection of 23 catchments, $C_Q$ compared to $C_{Pn}$ for catchments with FAR=S indicates a decrease in monthly
streamflow drought occurrence over time (Fig. 8, panel a, c, p, q, r, s, v and w). The decrease in streamflow drought occurrence is likely related to changes in reservoir outflow (example presented in S5) or to the construction of a reservoir during the period of record (example in S6). Furthermore, a multiyear impoundment period at the beginning of the period of record can be the cause of lesser drought months at the end of the record (S7). For catchments labelled with FAR=G, the number of streamflow drought months mostly increases over time (Fig. 8, panel b, f, h, k, m, n; an example is presented in
S8). However, some catchments with FAR-code G show an opposing pattern where the number of streamflow drought months decreases over time (Fig. 8, panel g, j, o and t), likely related to changes in management and abstraction policies (S9).

### 5 Discussion

#### 5.1 Interpretation of the impact of human influences

Storage and impounding reservoirs were constructed in the UK for various reasons including hydropower generation, water supply, and flood control (Acreman, 1999). Reservoir operations have a direct impact on streamflow and can completely change the flow regime (e.g., Acreman et al., 2009; Acreman., 2016; Richter et al., 1996). This study identified several streamflow records that are particularly impacted by storage or impounding reservoirs (FAR=S) that showed streamflow drought characteristics that deviate from those expected under natural conditions. Some of these catchments showed a
change in streamflow drought occurrence over time (fewer drought months towards the end of the record), the reason for this change being the construction of a reservoir in the middle of the record (example presented in S6). A similar impact has also been described downstream of the Santa Juana dam in Chile by (Rangecroft et al., 2016), where outflow from the constructed reservoir was used to support irrigation downstream. However, this mitigating effect of a reservoir (and thus increased flow) might not be directly visible after construction as further impounds may temporarily decrease flow and
intensify streamflow drought, as was shown for example downstream of the Three Gorges Dam in China (Dai et al., 2008). For similar reasons, a multiyear filling period of the Llyn Brenig reservoir, which has a storage of 3 years average runoff





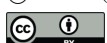

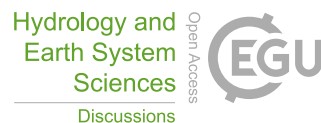

(Lambert, 1988), at the beginning of the record resulted in a large proportion of streamflow drought months (example shown in S7). In other cases, reservoirs were already present at the beginning of the record and temporal changes are likely to be related to changes in management practices (example in S5), in particular releases from impoundments to increase downstream flows. Overall, this study also found that reservoirs reduce the correlation between streamflow variability and

meteorological drought indices, especially in summer months. This is partly related to the use of reservoirs to compensate for low flows in the main branches of downstream rivers (for example, in the river Dee basin presented in S2). Similarly reduced correlations between meteorological drought indices and streamflow were also found for streamflow downstream of reservoirs on the Iberian Peninsula (Vicente-Serrano et al., 2014; López-Moreno et al., 2013).

Groundwater abstractions are made for, e.g., public water supply in various regions of the UK (mainly from the Chalk and

Permo-Triassic sandstones). Results of this study show that some of the catchments impacted by groundwater abstractions have streamflow droughts with longer average durations (example in S1). An increase in duration (compared to the modelled natural flow) has also been shown for the upper Guadiana River in Spain (Van Loon & Van Lanen, 2013). Furthermore, some of the identified catchments with groundwater abstractions show an increase in streamflow drought occurrence towards the end of the record (S8), possibly related to more intensified groundwater usage. However, the impact of groundwater

abstraction is not uniform and many records suggest that they do not necessarily result in prolonged streamflow drought duration or an increase in streamflow drought occurrence towards the end of the record. Moreover, groundwater management practices have increasingly focused on environmental problems related to low flows. Concerns over low flows which were partly caused by intensive abstraction, increasing in the late 1980s and early 1990s (NRA, 1993), resulted in a growing trend towards moderating abstraction, including schemes such as 'Alleviation of Low Flows', whereby 40 rivers

with problematic low flows were identified and (the feasibility of) different solutions, such as a reduction of abstraction or augmentation of flow with groundwater, were investigated and applied. One of the top 40 low flow rivers is the Darent (example in S9) which showed fewer streamflow drought months towards the end of the record after a peak mid-record, most likely related to the in 1993 proposed action plan that includes a reduction in abstraction amounts from sensitive boreholes (NRA, 1993).

**5.2 Suitability of deviating streamflow records for drought monitoring and early warning**

Besides identifying human induced deviations in the instream drought situation of a particular river, the adopted screening approach can also be beneficial to evaluate how well streamflow drought indices reflect the country or regional scale drought situation as employed in large scale monitoring and early warning systems, e.g. the UK National Hydrological Monitoring

Programme (https://nrfa.ceh.ac.uk/nhmp). In the case of temporal changes in streamflow drought occurrence due to, e.g., consistent changes in reservoir outflow (S5) or abstraction rates (S9) or the occurrence of significant human disturbances during the beginning of record (S7), a streamflow drought index might either consistently or rarely indicate drought conditions, which could be discordant with the overall drought impacts experienced on the ground in the region. Furthermore, in the case of low correlations between meteorology and streamflow (for example in the case of compensation



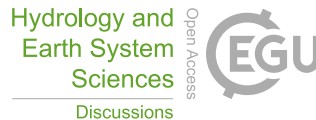

flows or water transfers), a streamflow index might indicate wet conditions during drought and vice versa, dry conditions during wet years. For example, the severe drought in the summer of 1984 in the river Dee basin (Lambert, 1988) was not indicated by part of its upstream stations (S2). Similarly, streamflow measured in a river used for water transfers in the South-eastern UK (S4) did not indicate drought conditions during the severe 1995-1997 drought (Marsh et al., 2007).

Furthermore, a streamflow drought index derived from a stream affected by groundwater augmentation (S3) would indicate the termination of 1976 drought in May, whereas this drought lasted until the end of summer (Marsh et al., 2007). These human influences, and the message that streamflow drought indices derived from these human influenced records provide, might be known to local water managers but may be unknown at large scale (e.g. to users of national to continental) drought monitoring and early warning systems, potentially leading to false alarm or misses of relevant drought conditions.

**5.3 Method and data caveats**

This study screened streamflow records for potential impacts of human influences on the streamflow drought signal and related these impacts to independent station metadata on human influences to provide evidence for consistency with these influences, or otherwise However, there may be other influences that are not reflected by either the used FAR-codes or station thumbnails that might affect streamflow and consequently streamflow drought characteristics. Examples include land

use (changes) such as urbanization (linked to, e.g., increased catchment responsiveness or over-exploitation of groundwater under cities (Schirmer et al., 2013) or changes in agricultural land use and cropping (e.g. Zhang & Schilling., 2006). Furthermore, England and Wales have well-developed drought management frameworks, and statutory drought response and management plans specifically targeting a reduction in water usage during drought events. Such factors have changed over time and may alter precipitation-flow relationships in complex ways that are not captured by the categorical FAR system and

descriptive station thumbnails used here.

For the screening of catchments with deviating streamflow drought characteristics, a dataset of catchments labelled with FAR=N (near-natural flow) was used to define a baseline for the relation between streamflow, precipitation and catchment characteristics expected in catchments with near-natural flow. The observed deviations of these relationships for influenced records are contingent on the quality of this baseline, which is clearly imperfect – issues include the spatial

representativeness of the 'N' catchments versus influenced catchments, the non-linear relationship between precipitation and flow (meaning deviations from this relationship may not scale in a linear way) and so on. However, it must be borne in mind that there is rarely ever a 'perfect' baseline using any approach: the natural condition is often simply not known (in an influenced catchment), there are always issues in extrapolation in paired catchments, and even when 'pre-impact' series are available climate variability is a confounding factor. Future research should aim at improving this baseline; for example, the

correlation between near-natural streamflow and meteorology might improve when evapotranspiration is considered (as is for example shown for the Iberian Peninsula by Vicente-Serrano et al., (2014)).

The screening approaches are not solely based on changes in the streamflow record, and also consider precipitation and catchment characteristics. Furthermore, deviations are related to FAR-codes and thumbnail station description. However, it







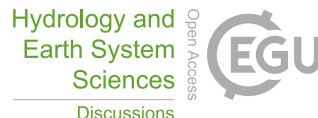

should be highlighted that this is not an attribution study. True attribution requires specific case study scale research. This is especially the case for the diffuse (indirect) impacts of groundwater abstractions on the increase in streamflow drought duration and occurrence over time that were found for, e.g., the Mimram Panshanger Park (S1) or the Cam at Dernford (S8). True attribution is beyond the scope of this large-scale (national) assessment, but an important topic for future research,

especially because these catchments are most sensitive to prolonged streamflow drought durations and an increase in streamflow drought occurrence over time. True attribution further requires more data about the type, degree and historical changes of human influences. Knowing that a catchment is impacted by groundwater abstraction is not sufficient as the overall impact of these groundwater abstractions could potentially be intensifying (S1 or S8), mitigating (S3), could have changed over time (S9) or is not detectable at all (Fig. 4 or 6). Such metadata sets – particularly on the degree of influence -

are rarely available, internationally. Even in the UK, which has a robust regulatory system and good data on water management practices, such data are often not readily accessible for research, or require a significant amount of processing to make them useable. While there are ongoing efforts to process such datasets and make them more available, any conclusions on degrees of influence are likely to remain subject to significant uncertainties.

To address the topic for more metadata, Van Loon et al., 2016a highlight the need for a bottom-up approach to collect more

data about human influences. While such a database would be a valuable research tool, it is a major challenge and effort to index the type and quantify the degree of all different human influences for each and every single catchment. Complementing the bottom up approach by the screening methodology applied here, i.e. using a large set streamflow records as a starting point to isolate influenced records with deviating streamflow drought characteristics, may help progress towards attributing or modelling these deviations. While the number of catchments in such a case-study dataset may be relatively

small (compared to the total number streamflow records available), it may allow for a more targeted collection and quantification of the range of human influences. Such an approach was advocated by Marsh (2002) who proposed that 'impact' catchments – with known influences, and a demonstrable effect of these influences in the record – should be an important counterpart to 'Benchmark' reference networks. Such networks would accelerate research to improve our understanding of how different human influences modify streamflow drought characteristics, alongside other flow regime

properties.

## 6 Conclusion

This study aimed to identify catchments with deviating streamflow drought characteristics using a dataset of streamflow records from England and Wales with indicative metadata on human influences on flow regimes, namely the 'Factors Affecting Runoff' codes from the National River Archive ("groundwater abstractions" and "storage or impoundments").

Some of the identified catchments affected by groundwater abstractions revealed prolonged streamflow drought durations. Furthermore, the distribution of streamflow drought occurrence over the period of the record revealed a decrease in streamflow drought months over time for some of the catchments with "storage or impounding reservoirs" and both an





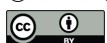

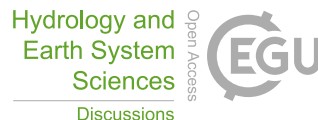

increase and decrease in monthly streamflow drought occurrence for some of the catchments labelled to be affected by "groundwater abstractions". The correlation between streamflow and precipitation was weaker for both catchments with "storage or impounding reservoirs" and "groundwater abstractions", respectively related to compensation flow and flow augmentation with groundwater during drought.

The change in streamflow drought occurrence over time and the diminished correlations between streamflow and precipitation affects the suitability of a streamflow drought index to reflect the overall drought situation. For example, anomalously high flow conditions due to compensation flow from reservoirs, water transfers or groundwater augmentation can occur during drought situations. Furthermore, non-stationarities in flow records, caused by, e.g., the filling or construction of a reservoir upstream of the gauging station, affects their suitability to reflect current drought conditions. The

screening approaches for temporal changes in streamflow drought occurrence and weaker correlations between streamflow and precipitation were shown to be successful in filtering out some records with large disturbances that are likely less suitable for the monitoring of the overall drought impacts. Since this screening approach is parsimonious - based only on river flow records, precipitation, simple catchment descriptors and a categorical presence/absence flag for human influences - it has the potential to be easily applied in other regions as a first order assessment, pending a more detailed appraisal of

human influences datasets.

Human influences do not have a consistent effect on the various streamflow drought characteristics. The same human influence may intensify or mitigate streamflow drought characteristics. Furthermore, the human influence might be minimal or have changed over time due to, e.g., river restoration programs or changes in minimum flow requirements. This variety highlights the importance of not only indexing information about the type of human influences, but also the degree, overall

effect (intensifying or mitigating) and its changes over time. Approaches that take the streamflow record as a starting point and screen for records with deviating streamflow drought characteristics could prove useful by creating a smaller subset of heavily influenced records for which a targeted collection of all different human influences, and research on their impact, is more feasible; an important next step towards a better understanding of streamflow drought propagation in a human-modified world.

**Competing interests:** The authors declare that they have no conflict of interest.

**Acknowledgments**

This study is an outcome of the Belmont Forum project DrIVER (Drought Impacts: Vulnerability thresholds in monitoring and Early warning Research). Funding to the project DrIVER by the German Research Foundation DFG under the

international Belmont Forum/G8HORC's Freshwater Security programme (project no. STA-632/2-1) is gratefully acknowledged. Financial support for J. Hannaford within the DrIVER project was provided by the UK Natural Environment Research Council (grant NE/L010038/1), with additional contribution from the project Historic Droughts (grant NE/L01016X/1). We thank the following CEH staff for providing data and advice: Katie Muchan, Maliko Tanguy and Oliver Swain. The use of streamflow and precipitation data as well as catchment descriptors that were provided by the





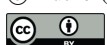

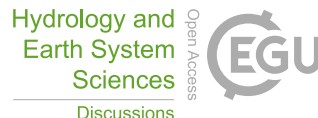

National River Flow Archive (NRFA), is gratefully acknowledged. The streamflow data and metadata used in this study is freely available at the NRFA website (http://nrfa.ceh.ac.uk/), and other datasets available on enquiry.





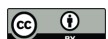

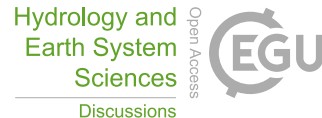

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





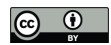

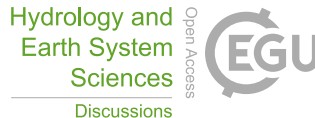

**Table 1**: FAR codes and their meaning.

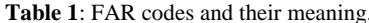

| FAR | Activity |
| --- | --- |
| S | Storage or impounding reservoir |
| R | Regulations |
| P | Abstractions for public water supply |
| G | Groundwater abstractions |
| I | Industrial and agricultural abstractions |
| E | Effluent return |
| H | Hydro-electric power generation |
| N | Near-natural flow records |





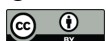



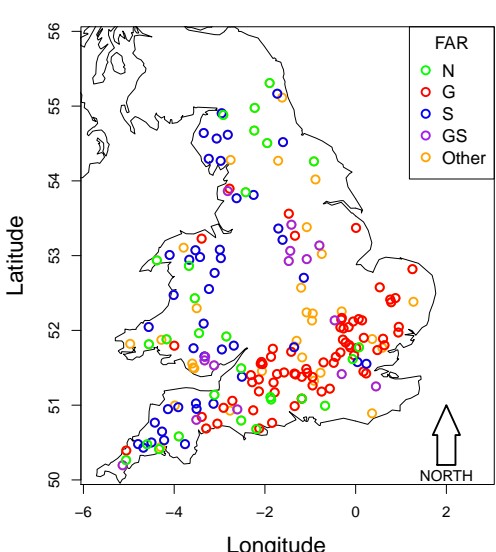

**Figure 1.** Gauging station locations and Factor Affecting Runoff (FAR) codes for corresponding catchments.





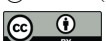

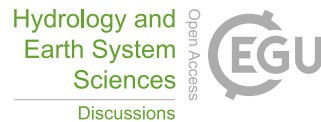

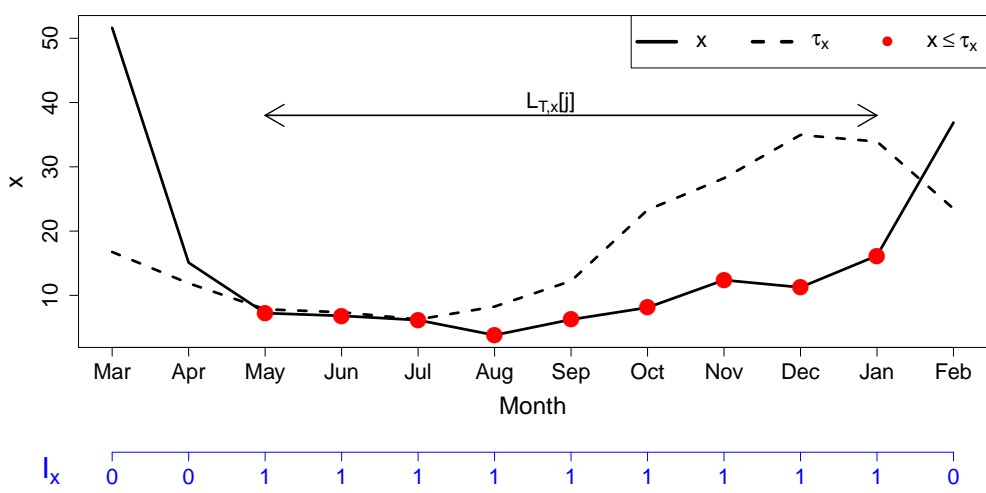

**Figure 2.** Schematic time series of variable x and corresponding drought threshold $\tau_x$ with derived duration of drought event j ($L_{T,x}$ [j]) and binary index time series of monthly drought occurrence ($I_x$).





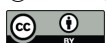

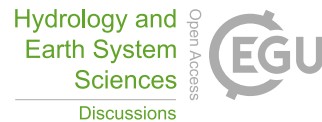

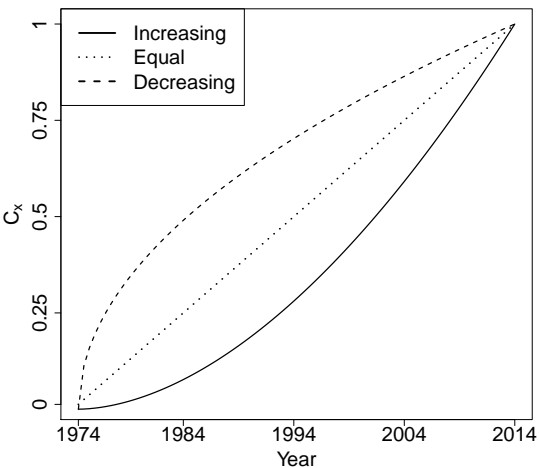

**Figure 3.** Schematic example of the evolution of the relative cumulative sum of monthly drought occurrence ($C_x$) for three cases: increasing number of drought months towards the end of record (solid line), a constant distribution of drought months over the period of record (dotted line), decreasing number drought months towards the end of record (dashed line).



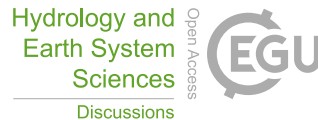

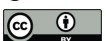

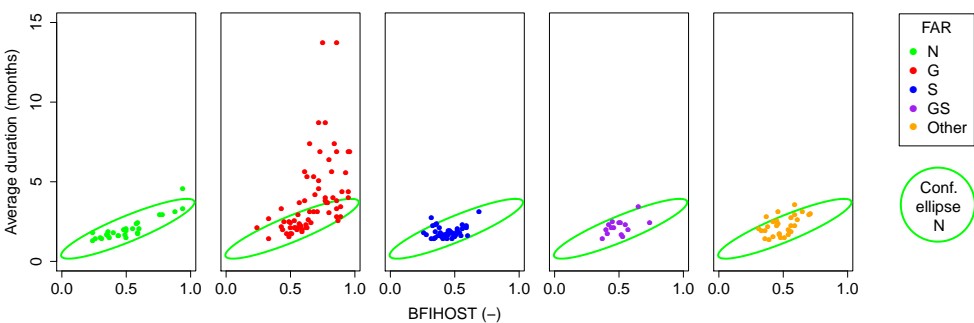

**Figure 4.** Average streamflow drought duration ($\overline{L_{T,Q}}$) versus BFIHOST for catchments labelled with different FAR codes (colours). Ellipse reflects is the 95% confidence ellipse for catchments with near-natural flow records (FAR=N).



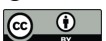


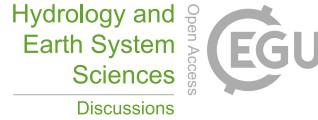

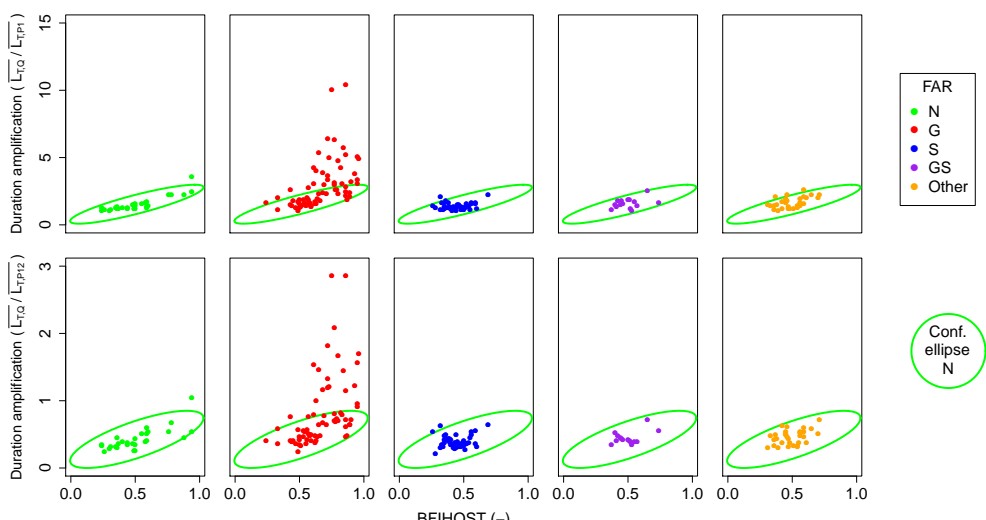

**Figure 5.** Amplification of average monthly streamflow drought duration over average monthly precipitation drought duration ($\overline{L_{T,Q}}/\overline{L_{T,P1}}$) (upper) and $\overline{L_{T,Q}}/\overline{L_{T,P12}}$) (lower) versus BFIHOST for catchments labelled with different FAR codes (colours). Ellipse reflects is the 95% confidence ellipse for catchments with near-natural flow records (FAR=N).



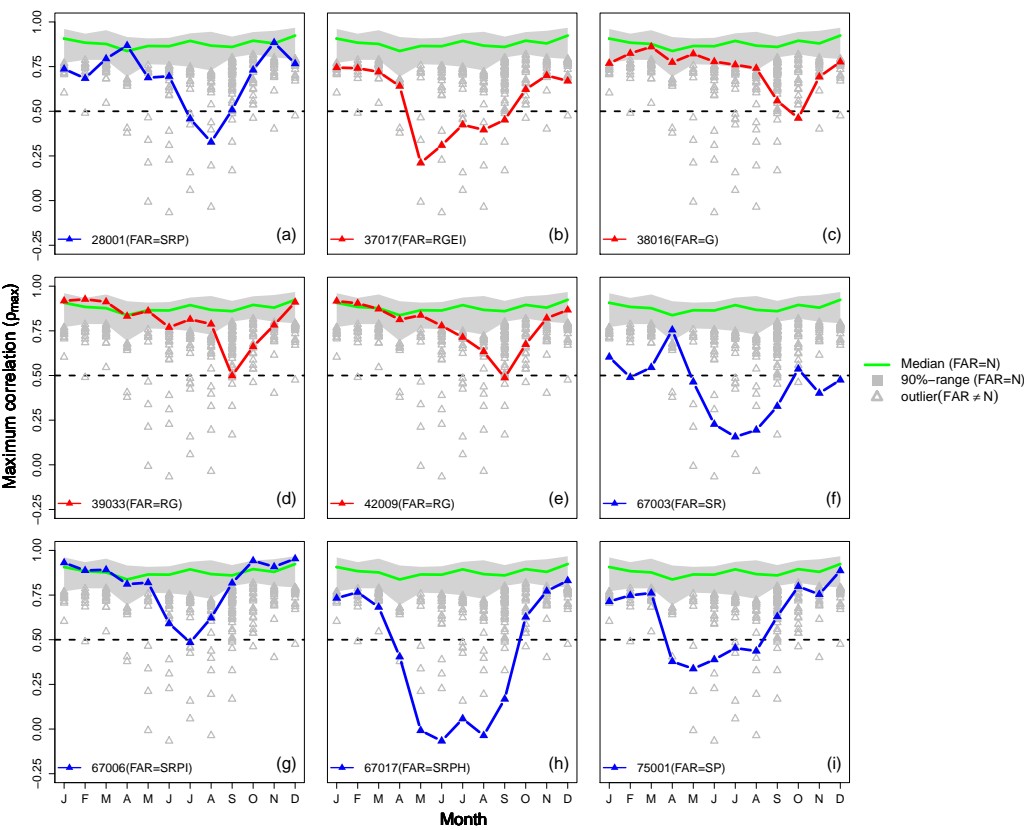

**Figure 6.** Maximum correlation ($\rho_{max}$) between streamflow (Q) and precipitation accumulated over different time periods (Pn) for catchments with near-natural flow (FAR=N). Grey triangles reflect $\rho_{max}$ of influenced catchments (FAR≠N) below the 90% range of catchments with near-natural flow. Coloured lines show $\rho_{max}$ of stations with at least one month $\rho_{max}<0.5$.





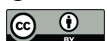

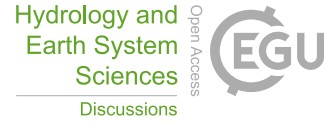

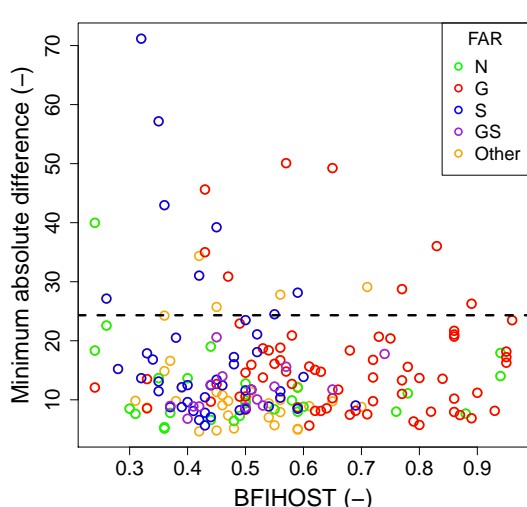

**Figure 7.** Minimum absolute difference ($A_{dif,min}$) between the relative cumulative sum of monthly streamflow ($C_Q$) and precipitation ($C_{Pn}$) drought occurrence versus the BFIHOST for records with different FAR. Dashed line indicates selected subgroup of catchments with strongest deviating timing in drought occurrence ($A_{dif,min} > 24.3$).





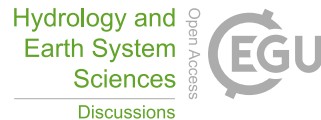

**Figure 8.** Relative cumulative sum of monthly drought occurrence ($C_x$) for strongly influenced records of streamflow ($C_Q$, coloured lines) and precipitation ($C_{Pn}$, grey lines). Plot titles indicate plot label (letter between brackets) and NRFA station name of each subplot.