# Peer review of "Human influences on streamflow drought characteristics in England and Wales"

_Hydrology and Earth System Sciences, 2017_

## Referee Comment (RC1) · Anonymous Referee #1 · 7 Sep 2017

Review of: Human influences on streamflow drought characteristics in England and Wales. This study analyses the impact of human perturbation of hydrological drought occurrence using streamflow and precipitation data.

The manuscript is very well written and organized, and introduction and discussion sections are really very sound. I consider the manuscript's topic is also highly suitable for HESS and it contains some novel issues, including the methodology applied to identify anthropogenic modifications of streamflow.

I would recommend the publication of the article in HESS. I include some suggestions and comments related to the need of including some clarifications in the methodological section:

[Figure]

**HESSD**

Page 2. 12. See also Vicente-Serrano et al. (2017) Journal of Hydrology:Regional Studies 12: 13-32, which is covering a similar topic.

Page 5.3 How were the monthly streamflow series created? Averaging the available daily records in a month?

Page 5.8 I understand the existing problems for data gap filling but the existence of gaps also limit calculation of drought indices. If I understand well, all the selected stations showed less than five days of missing data in all months between 1974-2013, so the entire monthly series were complete. If this is correct it should be stated in the manuscript.

Page 6.12-15. Why standardized streamflow and precipitation indices are not used instead of real precipitation and streamflow magnitudes? These indices are comparable spatially and seasonally. Note that streamflow and precipitation distributions are usually biased so this could have some impact on average precipitation and streamflow but also on total magnitude anomalies. The selection of this approach would be justified in some depth.

Page 9.21. It would be also quite interesting not only to analyse the magnitude of correlations but also the time-scales of precipitation accumulation that better correlates with streamflow. Maybe it could provide some relevant differences between natural and perturbed basins.

Page10.27. See also Vicente-Serrano et al. (2017) four further examples.

---

## Referee Comment (RC2) · Anonymous Referee #2 · 18 Sep 2017

Review Tijdeman et al. 2017 HESS

This manuscript nicely introduces an important topic, anthropogenic influences on streamflow drought, and outlines and investigates a core challenge when studying, identifying and monitoring streamflow droughts in the Anthropocene: human influences potentially changing the normal relationships, characteristics and descriptions of drought.

The paper is well written and fits the scope of HESS. Overall I believe that it gives a good outline of the issues involved in investigating the human influence, however there are a number of small changes which I believe should be addressed before acceptance.

[Figure]

The main changes that I believe are important are all listed below, and involve either some rewording, clarification in the methods section, small wording edits, and some suggestions for change or improvement on the figures. I think it is an important message and topic to address, however I think that the phrasing of the results and the main messages might need to be reconsidered slightly. Upon checking through the methods and figures of the paper, it raises the question of the length of all data that was used? Were all catchments analysed for the time period 1974 -2013, or did they vary in length (as the supplementary figures suggest). Furthermore, it is mentioned how many catchments were useable after the missing data criteria (187), however it would be useful to be more transparent about how many catchments were used for the analysis in this paper specifically. Figure 8 shows all 23 that are identified in Figure 7, however these 23 are not just the groundwater abstraction and storage FAR-codes that the paper has a focus on, and which are mentioned in the results section.

Please see below a list of the recommended changes.

Title: I would suggest a small edit to the title as you are not only looking at streamflow drought characteristics, but also at relationships. Perhaps this can be incorporated into the title if the authors agree that it is important too?

Abstract: Page 1, L16: You currently introduce BFI but this should be changed to introduce BFIHOST as you use this in the paper. L24: Change 'towards the end of record was found...' to 'towards the end of their records was found...' L26: Change 'screening approaches were successful' to 'screening approaches were sometimes successful' L30: Change 'approach' to 'approaches'

Introduction: Page2, L2: Change 'They are monitored...' to 'They are identified and monitored...' Page 3, L16: Change 'A first approach' to 'One suggested approach' L19 – 27: Please reword these sentences to be clearer, in particular lines 19-21 – do you mean 2 discharge stations or 1 discharge station and 2 time periods? L27: Change 'However, the above described approaches...' to 'However, all these approaches...'

L29: Suggested paragraph break at "An alternative" Page 4, L18-10: rephrase L12: Change 'here we propose a screening approach that seeks to' to 'here we propose screening approaches that seek to' given that you use three different approaches, in varying results. L15: Change 'This study used a diverse' to 'This study uses a diverse'

2. Study area and Data Page 5, L1: Introduce the abbreviation for NRFA here L8: 1973 – should that be 1974 as that is the date used elsewhere and on the plots? L12-15: You introduce BFIHOST, but it might be useful to explain that it runs from 0 to 1 and what these mean to help the reader interpret your figures (Fig. 3 and 4) later on. Page 6, L6: typo?: NRA., 1993

3. Methods Page 6, L13-15: be clearer about introducing the streamflow and then the precipitation data. A suggestion for this would be in insert 'For precipitation droughts' in front of 'monthly accumulated precipitation P(t)...' on line 14, to have that clear distinction to the reader. L15: Reference McKee et al. 1993 when you introduce SPI L19: Reference Yevjevich 1967 when you mention threshold level approach Equation 2: I would question if this is needed Page 7, L1: suggested break or slight rearrangement to separate 'The following two characteristics were used in the ...' from the text before. L5: I would recommend keeping point 2 as brief as point one, and finding a way to explain it separately like you did for the duration Page 8, L14: you mention the 5th quantile here, but elsewhere you mention the 95th quantile.

4. Results In general – look to relate this section more to your hypothesis, in the structuring and/or the statements you make. I also believe that you don't state enough that the results from the other human influences do not seem to be outside the N 95% confidence ellipse. Page 8, L7: change 'the remaining catchments...' to 'the remaining G catchments' L9: Look to add that the other FAR (other than G) fit within the N 95% confidence ellipse. Page 10, L7: after 24.3 add '(indicated on Fig. 7)'

5. Discussion L22-24: state which figure/figures show this. L27: typo: bracket before Rangecroft should actually be before 2016. Page 11, L22-24: interesting, do you have

any abstraction rate data that can help illustrate this point? Page 12, L4: add in the word 'meteorological' into 'severe 995-1997 drought'. L6: add 'meteorological' into '1976 drought' if that is what you are referring to. L9: Suggest ways forward here after mentioning this risk of false alarm or misses of relevant drought conditions? L31: typo: Vicente-Serrano et al. (2014) Page 13 L7-9: add in 'and different types of groundwater abstraction' as they can affect the impact. L14: typo: brackets missing for Van Loon et al. (2016a).

6. Conclusion L29: make into: 'National River Archive (focusing on "groundwater abstractions" and "storage or impoundments").'

Table 1: Suggestion to add some visual marker/flag to the FAR codes used in this study (N, S and G)

Figure 1 and 7: Suggestion to use different shapes as well as different colours to represent the different FAR-codes due to the consideration of black and white printing, if the authors deem this a good idea. [What is the spatial distribution spread of the BFIHOST categories? Just curious] Figure 4 and 5: the plots could be labelled on the top outside margin above each one as headings instead of needing a colour key to the right? Figure 5: Y axis could have clearer outside margin labelling – P1 and P12 Figure 6: Consider adding data points and data lines to legend Figure 7: Specify the use of the 95th quantile in the caption (you currently state the threshold value used, also useful) Figure 8: I would suggest moving the catchment codes to the left hand side of each plot to be closer to their labelling. You focus on FAR-codes G and S for most of the paper, but then in Figure 8 you include a whole series of different activities. Furthermore you only make reference to the G and S plots of Figure 8 in the text (section 4.3) so do you need all 23 plots?

Supplementary What I gather from the supplementary data is that a lot of the catchments have different periods of data availability, they were not all 1974 – 2013? If this is correct, that it might be worth making this clearer in the methods. Where the thresholds all calculated using the ∼40 years data, or were some done using much shorter time periods? (e.g. S2, S3, S4)

Overall, the supplementary data could be more obvious about the FAR-code at the start of each one. I recommend having the FAR-code with the current heading of catchment code for each catchment, e.g. S1 Mimram at Panshanger Park (38003) 38003 (FAR = GI)

Typos and comment through supplementary: S2: only 1 year S3: label the two catchments better S4: mismatch of dates. 1994-1998 in the plot but 1984 – 1998 in the caption/text S5: Add dates to figure caption S6: Add dates to figure caption, and add a horizontal line to indicate when the reservoir was build/ introduced into the catchment S8: Add dates. Y axis on top plot needs to be in full S9: change your location of legend to be consistent with S8. Y axis same as S8.

---

## Author Comment (AC1) · 5 Oct 2017

**Response to the comments of Anonymous Referee 1:**

Review of: Human influences on streamflow drought characteristics in England and Wales. This study analyses the impact of human perturbation of hydrological drought occurrence using streamflow and precipitation data. The manuscript is very well written and organized, and introduction and discussion sections are really very sound. I consider the manuscript's topic is also highly suitable for HESS and it contains some novel issues, including the methodology applied to identify anthropogenic modifications of streamflow.

We thank the reviewer for the positive and constructive feedback on the manuscript and are grateful for the valuable input. Below we respond (in blue) to the reviewer comments (which are in black).

Page 2. 12. See also Vicente-Serrano et al. (2017) Journal of Hydrology: Regional Studies 12: 13-32, which is covering a similar topic.

Thank you for making us aware of this publication, which we had missed.

Page 5.3 How were the monthly streamflow series created? Averaging the available daily records in a month?

Monthly streamflow time series were created by calculating the monthly average of all daily flows. We will clarify this in the revised version of the manuscript.

Page 5.8 I understand the existing problems for data gap filling but the existence of gaps also limit calculation of drought indices. If I understand well, all the selected stations showed less than five days of missing data in all months between 1974-2013, so the entire monthly series were complete. If this is correct it should be stated in the manuscript.

Streamflow records that had at least one month with more than five days of missing data were excluded from the analyses (so a record could have several months with a few days of missing data). We will clarify this in the revised version of the manuscript.

Page 6.12-15. Why standardized streamflow and precipitation indices are not used instead of real precipitation and streamflow magnitudes? These indices are comparable spatially and seasonally. Note that streamflow and precipitation distributions are usually biased so this could have some impact on average precipitation and streamflow but also on total magnitude anomalies. The selection of this approach would be justified in some depth.

For the correlation based screening approach (Results in section 4.2), we use rank correlation (Spearman's Rho). Thus, the strength of the correlation will be the same, regardless whether standardized or not. The other two screening approaches (results in section 4.1 and 4.3) are based on drought characteristics derived from a threshold based approach (20th percentile threshold). This is the classic, most used method to determine below-threshold characteristics and we think it an advantage to use such a threshold based approach on raw precipitation and streamflow data as it assures that the 20th percentile threshold is exceeded the expected 20 percent of the time for each station and calendar month.

This is not necessarily the case when streamflow data would be transformed to the Standardized Streamflow Index. An SSI record computed with for example the GEV distribution exceeds the 20th percentile (corresponding to an SSI value of -0.84) between 7.5 and 37.5 percent of the time, due to an imperfect distribution fit. Therefore, standardizing streamflow in combination with a threshold based approach reaches the opposite of having a fair comparison of a number of threshold exceedances over space and time.

Arguably, better results (a closer approximation of the 20[th] percentile of threshold exceedance) should be obtained using the best fitting distribution for each calendar month or station (as is done in Vicente-Serrano et al., 2012). However, even the "best fitting" distribution is likely not perfect and will result in a variability in threshold exceedances between stations and calendar months. This is especially the case for heavily influenced streamflow records that could have less standard distributions (e.g., bimodal in case of changes in outflow from a reservoir during the period of record). Concluding, we did consider using SSI, but decided against it as it adds unnecessary complexity to the interpretation effort due to the effects of distribution fitting, which would distract from the main topic.

Page 9.21. It would be also quite interesting not only to analyse the magnitude of correlations but also the time-scales of precipitation accumulation that better correlates with streamflow. Maybe it could provide some relevant differences between natural and perturbed basins.

For natural catchments in the UK, this has been done in (Barker et al., 2016); catchments with a high BFIHOST have a stronger correlation with the precipitation accumulated over longer timescales, as is also shown in Figure 1A of this reply. Similar patterns were observed for the subset of catchments for which groundwater abstractions (FAR=G) or reservoirs (FAR=S) are indicated (Figure 1B and 1C of this reply). However, some of the streamflow records for which groundwater abstractions have been indicated that have a lower BFIHOST show on average a stronger correlation with precipitation records accumulated over longer timescales (Figure 1B). The observed (non-linear) pattern looks very similar to the relation between BFIHOST and drought duration (Figure 4 of the manuscript); more persistent streamflow droughts of higher average durations goes together with a stronger correlation between streamflow and precipitation accumulated over longer time periods. For catchments for which storage or impoundment reservoirs have been indicated (FAR=S), there are a few stations with a low BFIHOST that have a stronger correlation with precipitation accumulated over longer time periods (Figure 1C).

Although interesting, we feel like that this analysis does not fit in within the current version of the manuscript but rather in a follow up study that explores these relations in more detail (more focused on drought propagation / drought monitoring and early warning).

[Figure]

**Figure 1.** Average accumulation period (in months) of the precipitation signal that has the strongest rank correlation with streamflow for all calendar months.

Page10.27. See also Vicente-Serrano et al. (2017) four further example

Thanks for this suggestion.

**References:**

Barker, L.J. et al., 2016. From meteorological to hydrological drought using standardised indicators. *Hydrol. Earth Syst. Sci.*, 20(6), pp.2483–2505. Available at: http://www.hydrol-earth-syst-sci-discuss.net/12/12827/2015/.

Vicente-Serrano, S.M. et al., 2012. Accurate Computation of a Streamflow Drought Index. *Journal of Hydrologic Engineering*, 17(2), pp.318–332.

---

## Author Comment (AC2) · 5 Oct 2017

**Response to the comments of Anonymous Referee 2**

This manuscript nicely introduces an important topic, anthropogenic influences on streamflow drought, and outlines and investigates a core challenge when studying, identifying and monitoring streamflow droughts in the Anthropocene: human influences potentially changing the normal relationships, characteristics and descriptions of drought. The paper is well written and fits the scope of HESS. Overall I believe that it gives a good outline of the issues involved in investigating the human influence, however there are a number of small changes which I believe should be addressed before acceptance.

We thank the reviewer for the elaborate and constructive feedback on the manuscript and are grateful for the valuable input. Below we respond (in blue) to the reviewer comments (which are in black).

Were all catchments analyzed for the time period 1974 -2013, or did they vary in length (as the supplementary figures suggest).

All catchments were analyzed for the time period 1974-2013. In the supplementary material, we zoom-in to some periods where the human influence was clearly visible. We will clarify this in the revised version of the manuscript and the supplementary material.

Furthermore, it is mentioned how many catchments were useable after the missing data criteria (187), however it would be useful to be more transparent about how many catchments were used for the analysis in this paper specifically.

We used all available NRFA streamflow records for England and Wales and removed those records that at the time of the analyses did not cover the complete time-span between 1974-2013 or had one or more months within this time span with five or more days of missing data. We will emphasize this in the revised version of the manuscript.

Figure 8 shows all 23 that are identified in Figure 7, however these 23 are not just the groundwater abstraction and storage FAR-codes that the paper has a focus on, and which are mentioned in the results section.

In the revised version of the manuscript, we will remove these "other" graphs from Figure 8 and mention that only the catchments with "G" and "S" are shown.

Title: I would suggest a small edit to the title as you are not only looking at streamflow drought characteristics, but also at relationships. Perhaps this can be incorporated into the title if the authors agree that it is important too?

Thanks for this suggestion. We considered using other titles that include words like propagation but in the end decided to use the broad term "characteristics", as the focus of the paper is on the screening approach and not really any specific drought (propagation) characteristics / processes. We therefore would like to leave the title as it is.

Page 1, L16: You currently introduce BFI but this should be changed to introduce BFIHOST as you use this in the paper.

We will change: "… the Base Flow Index, BFI (specifically: BFIHOST, the BFI predicted from the hydrological properties of soils)"

Page 1, L24: Change 'towards the end of record was found. . .' to 'towards the end of their records was found. . .'

We will change as suggested in the revised version of the manuscript.

Page 1, L26: Change 'screening approaches were successful' to 'screening approaches were sometimes successful'

We will change as suggested in the revised version of the manuscript.

Page 1, L30: Change 'approach' to 'approaches'

We will change as suggested in the revised version of the manuscript.

Page2, L2: Change 'They are monitored. . .' to 'They are identified and monitored. . .',

We will change as suggested in the revised version of the manuscript.

Page 3, L16: Change 'A first approach' to 'One suggested approach' L19 – 27:

We will change as suggested in the revised version of the manuscript.

Page 3, L 19-21 – do you mean 2 discharge stations or 1 discharge station and 2 time periods?

We mean one streamflow record. We will emphasize this by changing. "… on a comparison between the influenced and non-influenced part of the record for a particular location." to "… on a comparison between the influenced and non-influenced part of one particular streamflow record."

Page 3, L27: Change 'However, the above described approaches. . .' to 'However, all these approaches. . .'

We will change as suggested in the revised version of the manuscript.

Page 3, L29: Suggested paragraph break at "An alternative"

We will split the paragraph at the suggested point in the revised version of the manuscript.

Page 4, L8-10: rephrase

"This study aims to close this gap, e.g following the recommendation of Barker et al. (2016). It seeks to understand the human influences on streamflow droughts in England and Wales which are densely populated regions with a long settlement history and thus prevalent human influences on river flow."

Rephrase L12: Change 'here we propose a screening approach that seeks to' to 'here we propose screening approaches that seek to' given that you use three different approaches, in varying results.

We will change as suggested in the revised version of the manuscript.

Page 4, L15: Change 'This study used a diverse' to 'This study uses a diverse'

We will change this as suggested in the revised version of the manuscript.

Page 5, L1: Introduce the abbreviation for NRFA here

We will add the abbreviation to the full name of the National River Flow Archive written out in this sentence.

Page 5, L8: 1973– should that be 1974 as that is the date used elsewhere and on the plots?

1973 is correct as for the calculation of, e.g., the precipitation accumulated over 12 months (P12), precipitation of the current month and 11 previous months are needed. Thus, P12 of January 1974 is based on data from 1973.

L12-15: You introduce BFIHOST, but it might be useful to explain that it runs from 0 to 1 and what these mean to help the reader interpret your figures (Fig. 3 and 4) later on.

Thanks for this suggestion. We will add more explanation, including also the suggested ranges and a more detailed explanation/interpretation of these ranges, to the text in the revised version of the manuscript.

Page 6, L6: typo?: NRA., 1993.

No typo as the NRA refers to the "National Rivers Authority"

L13-15: be clearer about introducing the streamflow and then the precipitation data. A suggestion for this would be in insert 'For precipitation droughts' in front of 'monthly accumulated precipitation P(t). . .' on line 14, to have that clear distinction to the reader.

We will change as suggested to have a clearer distinction between streamflow and precipitation droughts.

L15: Reference McKee et al. 1993 when you introduce SPI

Thanks for this comment; we indeed missed some key references here. We will refer to McKee et al. (1993) in the revised version of the manuscript.

L19: Reference Yevjevich 1967 when you mention threshold level approach

We will refer to Yevjevich 1967 (and Zelenhasić & Salvai, 1987) in the revised version of the manuscript.

Equation 2: I would question if this is needed

We prefer to keep equation 2 as this is in line with Tallaksen et al. (2009).

Page 7, L1: suggested break or slight rearrangement to separate 'The following two characteristics were used in the . . .' from the text before.

We will add the suggested break in the revised version of the manuscript.

L5: I would recommend keeping point 2 as brief as point one, and finding a way to explain it separately like you did for the duration.

We will explain the cumulative sum of drought occurrence in the paragraph prior to the list and shorten bullet point 2.

In general – look to relate this section more to your hypothesis, in the structuring and/or the statements you make.

Thanks for this suggestion. In the revised version of the manuscript we will start each subsection (4.1, 4.2, 4.3) by repeating/linking to the specific hypothesis.

I also believe that you don't state enough that the results from the other human influences do not seem to be outside the N 95% confidence ellipse. & Look to add that the other FAR (other than G) fit within the N 95% confidence ellipse.

We will mention/emphasize on these results to the text of the results section in the revised version of the manuscript.

Page 8, L7: change 'the remaining catchments. . .' to 'the remaining G catchments'

We will change 'the remaining catchments. . .' to the 'the remaining catchments labeled with FAR=G. . .' in the revised version of the manuscript.

Page 10, L7: after 24.3 add '(indicated on Fig. 7)'

We will add: '(indicated in Fig. 7)' in the revised version of the manuscript

L22-24: state which figure/figures show this

We will state the corresponding Figures in the suggested sentence. We will further add a link to these figures in the following sentences of this discussion section.

Page 10, L24-25 "Some of these catchments showed a change in streamflow drought occurrence over time (fewer drought months towards the end of the record, Figure 8), …"

Page 11, L4-5: "Overall, this study also found that reservoirs reduce the correlation between streamflow variability and meteorological drought indices, especially in summer months (Fig. 6)."

L27: typo: bracket before Rangecroft should actually be before 2016.

We will correct this typo in the revised version of the manuscript.

L22-24: interesting, do you have any abstraction rate data that can help illustrate this point?

Although we share the opinion of the reviewer that this is an interesting point, we think that including such detailed information is outside the scope of the current study. Like any other catchments, we do not look at the specific changes in, e.g., abstraction schemes, groundwater levels or reservoir operations and purely use the indicative Factor Affecting Runoff codes and station descriptions. Therefore, we state later on in the discussion the importance of more case study scale research and that "True attribution requires specific case study scale research." (Page 13, L1). The Darent (to which the reviewer comment is related), or any other of the identified catchments with deviating drought characteristics, would be a good starting point for such detailed case studies.

We will, however, more elaborately describe the low flow alleviation measures taken for the Darent in the discussion section, including some references where these alleviation measures are described (e.g., NRA, 1993).

Page 12, L4: add in the word 'meteorological' into 'severe 995-1997 drought' & L6: add 'meteorological' into '1976 drought' if that is what you are referring to

1976 and 1995-1997 were not purely meteorological as many surface and subsurface water systems were affected. Therefore, we prefer not to specify and thus limit to 'meteorological' in the revised version of the manuscript.

Page 12, L9: Suggest ways forward here after mentioning this risk of false alarm or misses of relevant drought conditions?

We will add the following sentence to the end of Section 5.2:
"Therefore, the capacity of streamflow drought indices derived from heavily influenced records in reflecting the overall impacts of drought should be evaluated in large-scale drought monitoring systems"

Page 12, L31: typo: Vicente-Serrano et al. (2014)

We will correct the typo in the revised version of the manuscript.

Page 13 L7-9: add in 'and different types of groundwater abstraction' as they can affect the impact.

We will change: "of these groundwater abstractions" with "different types of groundwater abstraction" on Page 13, L8

Page 13, L14: typo: brackets missing for Van Loon et al. (2016a).

We will correct this typo in the revised version of the manuscript.

Page 13 L29: make into: 'National River Archive (focusing on "groundwater abstractions" and "storage or impounds").

We will change as suggested in the revised version of the manuscript

Table 1: Suggestion to add some visual marker/flag to the FAR codes used in this study (N, S and G)

In the revised version of the manuscript, we will add a visual marker (*) to the table and refer to it in the caption of the table.

Figure 1 and 7: Suggestion to use different shapes as well as different colours to represent the different FAR-codes due to the consideration of black and white printing, if the authors deem this a good idea. [What is the spatial distribution spread of the BFIHOST categories? Just curious]

We will apply the suggested change to the Figure.

Figure 4 and 5: the plots could be labelled on the top outside margin above each one as headings instead of needing a colour key to the right?

We would like to keep the color coding to make it consistent with other graphs. We will add descriptive labels to the headers of the other graphs in the revised version of the manuscript.

Figure 5: Y axis could have clearer outside margin labelling – P1 and P12

We will emphasize on the Y-axis that the first row is P1 and the second row is P12

Figure 6: Consider adding data points and data lines to legend

We will improve the legend

Figure 7: Specify the use of the 95th quantile in the caption (you currently state the threshold value used, also useful)

We will specify the use of the 95$^{th}$ percentile in the caption in the revised version of the manuscript.

Figure 8: I would suggest moving the catchment codes to the left hand side of each plot to be closer to their labelling.

We will move the labels as suggested in the revised version of the manuscript

You focus on FAR-codes G and S for most of the paper, but then in Figure 8 you include a whole series of different activities. Furthermore you only make reference to the G and S plots of Figure 8 in the text (section 4.3) so do you need all 23 plots?

We will remove these "other" graphs from Figure 8 and mention that only the catchments with "G" and "S" are shown in the revised version of the manuscript.

Supplementary What I gather from the supplementary data is that a lot of the catchments have different periods of data availability, they were not all 1974 – 2013? If this is correct, that it might be worth making this clearer in the methods. Where the thresholds all calculated using the ~40 years data, or were some done using much shorter time periods? (e.g. S2, S3, S4)

All catchments have 40 years of streamflow data between 1974-2013. In the supplementary material for some catchments (e.g. S2, S3, S4) we zoom in to time periods where the human influence was clearly visible. We will state this in the captions of the supplementary Figures that we zoom in to a specific time period and emphasize in the main text that each record has a 40 year length between 1974 – 2013.

Overall, the supplementary data could be more obvious about the FAR-code at the start of each one. I recommend having the FAR-code with the current heading of catchment code for each catchment, e.g. S1 Mimram at Panshanger Park (38003) 38003 (FAR = GI)

We will add FAR codes to the title of the plots in the revised version of the supplementary material

S2: only 1 year

We will mention that S2 is zoomed-in to one particular year of the 40 year period of record.

S3: label the two catchments better

We will add FAR codes to the plot labels.

S4: mismatch of dates. 1994-1998 in the plot but 1984 – 1998 in the caption/text

The caption of Figure S4 contains a typo and will be corrected in the revised version of the supplementary material.

S5: Add dates to figure caption

We will add the dates to the caption of the Figure S5.

S6: Add dates to figure caption, and add a horizontal line to indicate when the reservoir was build/ introduced into the catchment.

We will add dates to the capture of the Figure and the suggested line to the Figure to indicate when the reservoir was constructed in the revised version of the Supplementary material.

S8: Add dates. Y axis on top plot needs to be in full.

We will add dates to the Figure caption and change Y-axis.

S9: change your location of legend to be consistent with S8. Y-axis same as S8.

We deliberately placed the legend of Figure S9 in the right corner as it will cover the time series in when placed in the left corner. Therefore, we propose not to change the location of the legend. We will change the Y-axis as proposed in the revised version of the supplementary material.

**References**

McKee, T.B., Doesken, N.J. & Kleist, J., 1993. The relationship of drought frequency and duration to time scales. AMS 8th Conference on Applied Climatology, (January), pp.179–184. Available at: http://ccc.atmos.colostate.edu/relationshipofdroughtfrequency.pdf.

NRA, 1993. Low Flows and Water Resources. Facts on the Top 40 Low Flow Rivers in England and Wales, National Rivers Authority, Bristol

Tallaksen, L.M., Hisdal, H. & Lanen, H.A.J. Van, 2009. Space-time modelling of catchment scale drought characteristics. Journal of Hydrology, 375(3-4), pp.363–372.

Zelenhasić, E. & Salvai, A., 1987. A method of streamflow drought analysis. Water Resources Research, 23(1), pp.156–168.

---

## Author Response (AR1)

Dear Editor,

We uploaded a new version of the manuscript with the changes made as suggested in the response to the reviewers. The manuscript below is the one with track-changes of the changes in the main text. The new Figures (changed according to our response to the reviewers) can be found in the version of the manuscript without track changes. We also uploaded a new version of the supplementary material, revised according to the response to anonymous reviewer 2.

Kind regards,

Erik Tijdeman (on behalf of the co-authors)

[revised manuscript text omitted]